# Construction and Performance Monitoring of Innovative Ultra-High-Performance Concrete Bridge

Haena Kim [1], Byungkyu Moon [1], Xinyu Hu [1], Hosin (David) Lee [1,*], Gum-Sung Ryu [2], Kyung-Taek Koh [2], Changbin Joh [2], Byung-Suk Kim [2] and Brian Keierleber [3]

1   Iowa Technology Institute, University of Iowa, Iowa City, IA 52242, USA; haena612@gmail.com (H.K.); byungkyu-moon@uiowa.edu (B.M.); xinyu-hu@uiowa.edu (X.H.)
2   Department of Infrastructure Safety Research, Korea Institute of Civil Engineering and Building Technology, Goyang, Gyeonggi 10223, Korea; ryu0505@kict.re.kr (G.-S.R.); ktgo@kict.re.kr (K.-T.K.); cjoh@kict.re.kr (C.J.); bskim@kict.re.kr (B.-S.K.)
3   Buchanan County, 1511 First Street Independence, Iowa City, IA 50644, USA; bkeierleber@co.buchanan.ia.us
*   Correspondence: hosin-lee@uiowa.edu; Tel.: +1-801-512-4202

**Abstract:** The application of Ultra-High-Performance Concrete (UHPC) materials in rehabilitating bridges and constructing primary bridge components is increasing rapidly across the world because of their superior strength and durability characteristics when compared to regular concretes. However, there have been few new bridges constructed using UHPC materials with regular formworks, ready-mix trucks, and construction equipment. This paper presents a comprehensive report encompassing the design, construction, and performance monitoring of a new bridge constructed in Iowa using a unique UHPC technology that includes steel fibers of two different lengths embedded in the concrete. By using optimized lengths of steel fibers, both the tensile strength and the toughness were increased. The UHPC material was produced with local cement and aggregates in the US using typical ready-mix concrete equipment. This paper discusses the experience gained from the design and construction process including mix design, batching, delivery of steel fibers to the ready-mix concrete batch unit, and post-tensioning of precast slabs at the jobsite. For four years after construction, the joints of the bridge decks were monitored using strain sensors mounted on both sides of the deck joints. The strain values were quite similar between the two sides of each joint, indicating a good load transfer between precast bridge girders. A bridge was successfully constructed using a unique UHPC technology incorporating two different lengths of steel fibers and utilizing local cement and aggregates and a ready-mix truck, and has been performing satisfactorily with a good load transfer across post-tensioned precast girder joints.

**Keywords:** UHPC; strain gauge; field construction; post-tensioning; monitoring; bridge; joint; steel fibers



## 1. Introduction

Concrete is the most commonly used building material in the world. A total of 4.1 billion tons of cement were used in 2019, contributing approximately 7% of global $CO_2$ emissions, assuming 0.9 pounds of $CO_2$ generated for the manufacturing of 1 pound of cement [1]. The proposed Ultra-High-Performance Concrete (UHPC)'s significantly higher strength allows for the use of a slender member with smaller cross-sections and with a longer service life, which will reduce the amount of $CO_2$ emissions associated with building concrete bridges [2].

Concrete technology has evolved through the optimization of mix ingredients to improve strength, workability, and durability. In the 1980s, High-Performance Concretes (HPCs) were developed, with significantly improved durability and compressive strengths ranging from 48 to 117 MPa [3]. In the 1990s, UHPC was introduced in France and the first UHPC pedestrian bridge, with a span of 60 m, was built in Quebec, Canada [4]. The first

UHPC bridge in the US was constructed in 2006 in Wappello County, Iowa, and another UHPC bridge was built in Buchanan County, Iowa, in 2008 [5,6].

UHPC is a dense flowable mix which is mainly composed of fine masonry sands (no coarse aggregates) and steel fibers with a high-range water reducer to produce a very low water/cement ratio of around 0.25. UHPC typically exhibits a compressive strength greater than 150 MPa (21.7 ksi) and a sustained post-cracking tensile strength greater than 5 MPa (0.72 ksi) [7]. It is common to include a high proportion of silica fume up to 10 percent relative to the weight of cement. UHPC displays a discontinuous pore structure that enhances its durability. The most common steel fiber used in UHPC constructions is a 0.2 mm diameter by 13 mm long straight fiber with a specified minimum tensile strength of 2000 MPa (290 ksi) [8].

This paper presents the laboratory test results, design, construction, and monitoring of a new bridge using a unique UHPC technology which utilizes two different lengths of steel fibers, 16.3 mm and 19.5 mm, at a 1:2 ratio by weight. Lengths of steel fibers were optimized to increase both the tensile strength and the toughness. This unique UHPC mix was used to build a pi-girder bridge in Buchanan County, Iowa. After the bridge was constructed, the bridge joints were monitored three times over four years using strain sensors attached on both sides of each joint.

## 2. Background

Over the past 15 years, as shown in Figure 1, the number of UHPC projects completed each year has been steadily increasing [9]. As depicted in Figure 2, UHPC has been applied in over 250 bridges in the US, mainly as precast concrete deck panels and the composite connections between precast deck panels and supporting girders [9]. Until recently, UHPC materials were predominantly used for field-cast connections between prefabricated bridge elements [10]. Steel fibers cost $615/m$^3$ when they are added at 1.5% by volume, which represents about half of the total cost of UHPC at $1110/m$^3$ [3]. Due to the high cost of steel fibers in UHPC materials, UHPC has been more commonly used as a deck overlay rather than for building new bridges [11]. As a result, to date, only three bridges have been constructed in the US using UHPC materials, all in Iowa. An increased tensile bond strength of a specially formulated UHPC overlay on existing bridge decks has been reported [12].

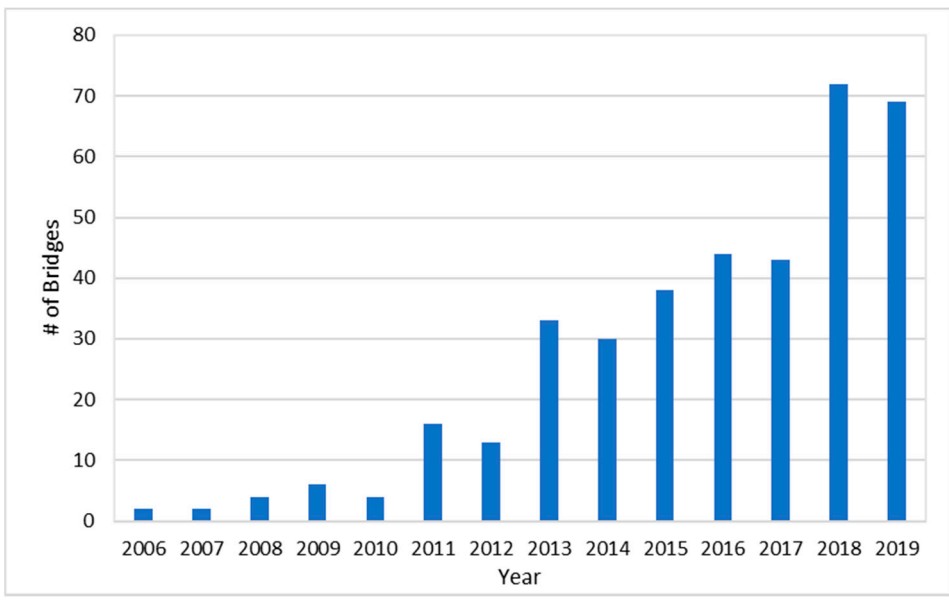

**Figure 1.** Number of UHPC projects each year [9].

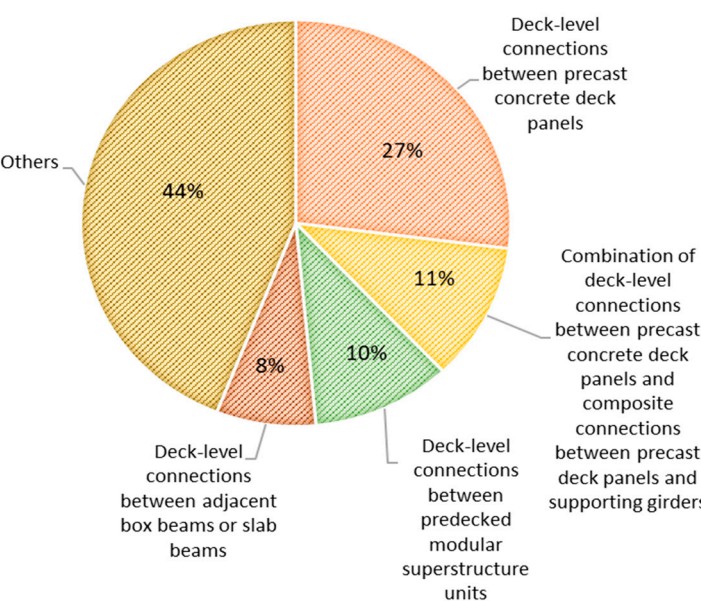

**Figure 2.** Use of UHPC materials in bridge components in the United States [9].

Grouts have a significant impact on a bridge's deck-level connection performance. To improve the structural performance of the connections between bridge decks, UHPC grout with a lower shrinkage and enhanced bonding strength has been adopted [13]. Due to UHPC grouting material's improved shear bearing capacity, the beam stress is effectively reduced and the development of oblique cracks is inhibited [14]. To better comprehend the behavior of precast deck connections for accelerated construction, a new connection detail to connect a precast column to a cap beam was analyzed using nonlinear finite element analysis [15]. Furthermore, small-scale direct shear tests and large-scale double shear push-off tests were performed on innovative connections constructed using UHPC materials. The novel connection details were reported to have the potential to meet the existing strength limit requirements outlined in the AASHTO bridge design specification [16].

UHPC does not exhibit a significant amount of drying shrinkage because a large amount of steel fibers in the HPC help reduce the shrinkage cracking and redistribute the shrinkage strains [17]. UHPC materials have less propensity for shrinkage cracking and shrinkage cracking occurs in the form of microcracks due to the steel fibers [18]. For the creep and shrinkage models for AASHTO LRFD, UHPC materials provide superior mechanical properties, a low water content, and a high volume of fiber reinforcement, which can enhance creep and shrinkage behavior compared to conventional concretes [19]. Full-depth application of UHPC at the punching shear area in flat slabs transferred the failure mode from a brittle punching shear failure to a ductile punching shear-flexural failure and improved both cracking strength and punching shear strength [20].

It is possible to construct a 300 foot (91.4 m) UHPC girder with a weight per unit length that is similar to that of an existing 200 feet (61 m) long conventional concrete girder [21]. This innovative slender and longer girder design will provide aesthetic value and more of the open space that is critically needed for smart cities. The adoption of UHPC materials in rehabilitating bridges and constructing primary bridge components is progressing rapidly in the world because of UHPC's unique characteristics of higher strength and longer durability than regular concretes [22].

## 3. Materials and Specimen Preparation

Table 1 summarizes the optimum mix design for the proposed UHPC material, which was developed by performing compressive and indirect tensile tests of numerous specimens. Short steel fibers prevent microcracks whereas long fibers prevent macro-cracks through their bridging effect [9]. Our unique UHPC material contained longer steel fibers

with lengths of 0.63 in (16.3 mm) and 0.78 in (19.5 mm) mixed at a respective ratio of 1.6% and 3.2% by weight, and they significantly increased flexural tensile strength. The water to cement (w/c) ratio of the UHPC was 0.23, which is the lowest amount of water needed for cement hydration. To improve the workability, based on the observations of the dry mixing condition, a slightly higher amount of superplasticizer (0.7% by weight) was used.

**Table 1.** Mix design of UHPC.

| Constituents | lb/yd$^3$ (kg/m$^3$) | lb/ft$^3$ | Percent by Weight |
|---|---|---|---|
| Sand | 1462 (867) | 54.1 | 35.30% |
| Cement | 1329 (789) | 45.9 | 32.10% |
| Water | 311 (184) | 11.5 | 7.50% |
| Superplasticizer | 31 (18) | 1.15 | 0.70% |
| 16.3 mm fiber | 66 (39) | 2.4 | 1.60% |
| 19.5 mm fiber | 131 (78) | 4.8 | 3.20% |
| Defoamer | 1 (0.5) | 0.04 | 0.02% |
| Shrinkage-reducing agent | 13 (8) | 0.48 | 0.30% |
| Premix * | 797 (473) | 29.5 | 19.30% |
| Total | 4142 (2457) | 153.4 | 100% |

* Silica fume, ground quartz, and other performance enhancers.

First, all dry materials, namely cement, sand, and premix, were mixed in a 2-cubic concrete mixer at a speed of 20 rpm for 5 min. The water and other liquid additives such as superplasticizer and defoamer were then added and mixed for an additional 4 min. Finally, after the mix was evenly mixed, steel fibers were added to produce the laboratory mix. During the mixing process, extra amounts of water and superplasticizer were added to improve workability of the mix. For example, 0.9 kg of extra water and 0.74 kg of superplasticizer were added to the laboratory mix, resulting in higher w/c ratios (to 0.27 from the original w/c ratio of 0.23).

*3.1. Compressive Strengths*

Both laboratory and field mix 3× 6 inch (7.6 × 15.2 cm, diameter × height) cylindrical specimens were tested for compressive strength using an Instron Prism 5500 (1.1 MN Capacity). The test results of both laboratory and field mixes are summarized in Table 2. As can be seen from plots of average compressive strengths with standard deviations shown in Figure 3, the wet-cured laboratory mix satisfied the target compressive strength of 180 MPa after 28 days, whereas the field mix slightly missed the target. As shown in Figure 3, the wet-cured laboratory mix achieved a very significant early gain in compressive strength of 87.9 MPa after one day, continued to gain strength until 14 days, and remained at a similar level after 28 days. The wet-cured field specimens exhibited similar levels of strength gain as those of wet-cured laboratory mix.

**Table 2.** Compressive strength and standard deviation results of lab and field mix specimens.

| Batch | Average and Standard Deviation of Compression Strength in MPa | | | | | | |
|---|---|---|---|---|---|---|---|
| | Day 1 | Day 2 | Day 4 | Day 7 | Day 14 | Day 18 | Day 28 |
| Lab mix (1 cf) | 87.9/10.8 | 114.3/15.4 | 133/3.2 | 155.8/3.4 | 174.4/5.3 | n.a. | 180/3.7 |
| Field mix (5.5 cy) | n.a. | n.a. | n.a. | 154.5/8.4 | 144/2.5 | 181/5.0 | 174.2/21.9 |

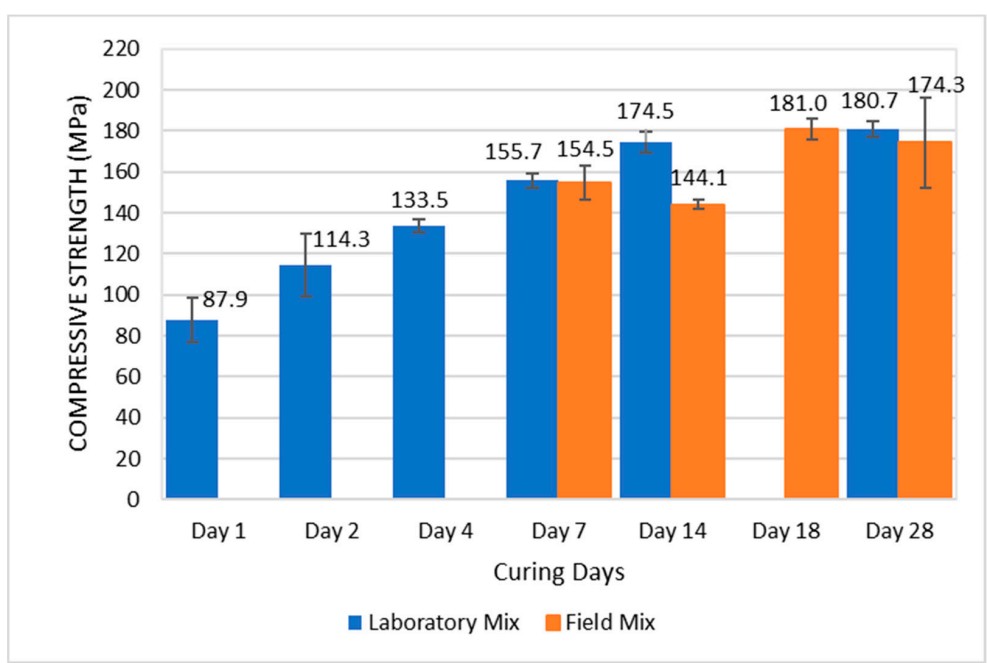

**Figure 3.** Averages and standard deviations of compressive strengths of laboratory and field mixes for 66 UHPC specimens.

*3.2. Indirect Tensile Strength*

For the laboratory mix (not the field mix) 3 by 6 in (7.6 × 15.2 cm, diameter × height) cylindrical specimens were prepared for indirect tensile strength testing. The average indirect tensile strengths and standard deviations are plotted in Figure 4. The wet-cured laboratory mix specimens achieved the highest indirect tensile strength of 18.3 MPa after 7 days and maintained the same strength after 14 days.

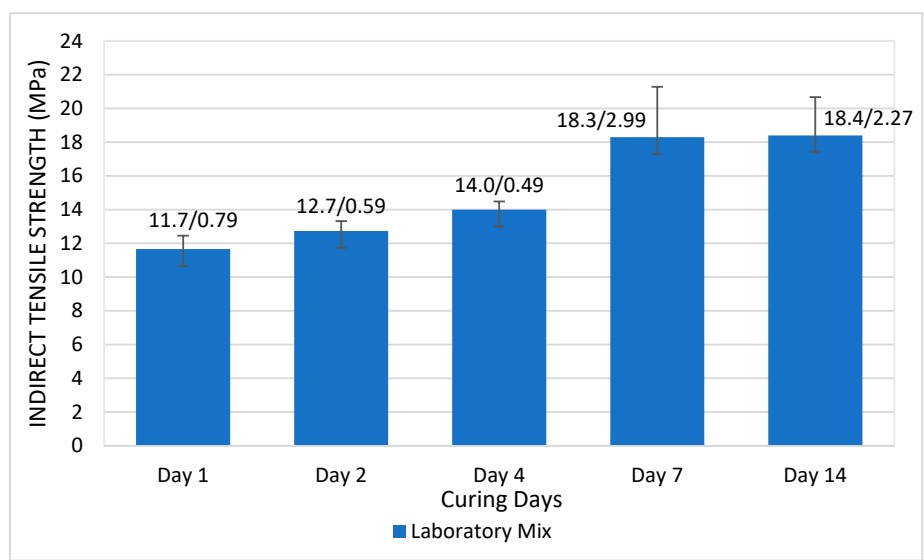

**Figure 4.** Averages and standard deviations of indirect tensile strengths of laboratory mixes.

## 4. Design of the Hawkeye UHPC Bridge

As shown in Figure 5, the Hawkeye Bridge is 52′ (15.8 m) long and 32′-5″ (10 m) wide with an innovative pi-girder design, where each girder is 52′ (15.8 m) long, 5′-3″ (1.6 m) wide, and 2′-4″ (0.7 m) deep. The Hawkeye Bridge was designed using the load factors and load combinations of the AASHTO-LRFD standard. Service load stress checking was applied to precast girders and ultimate strength design was applied to the girder deck and

abutment. Each of six girders was post-tensioned. Five crossbeams were installed and post-tensioned at 12′-9″ (3.9 m) spacing to ensure a proper lateral load distribution. The shear bulbs were then filled with UHPC materials to improve the connectivity between girders.

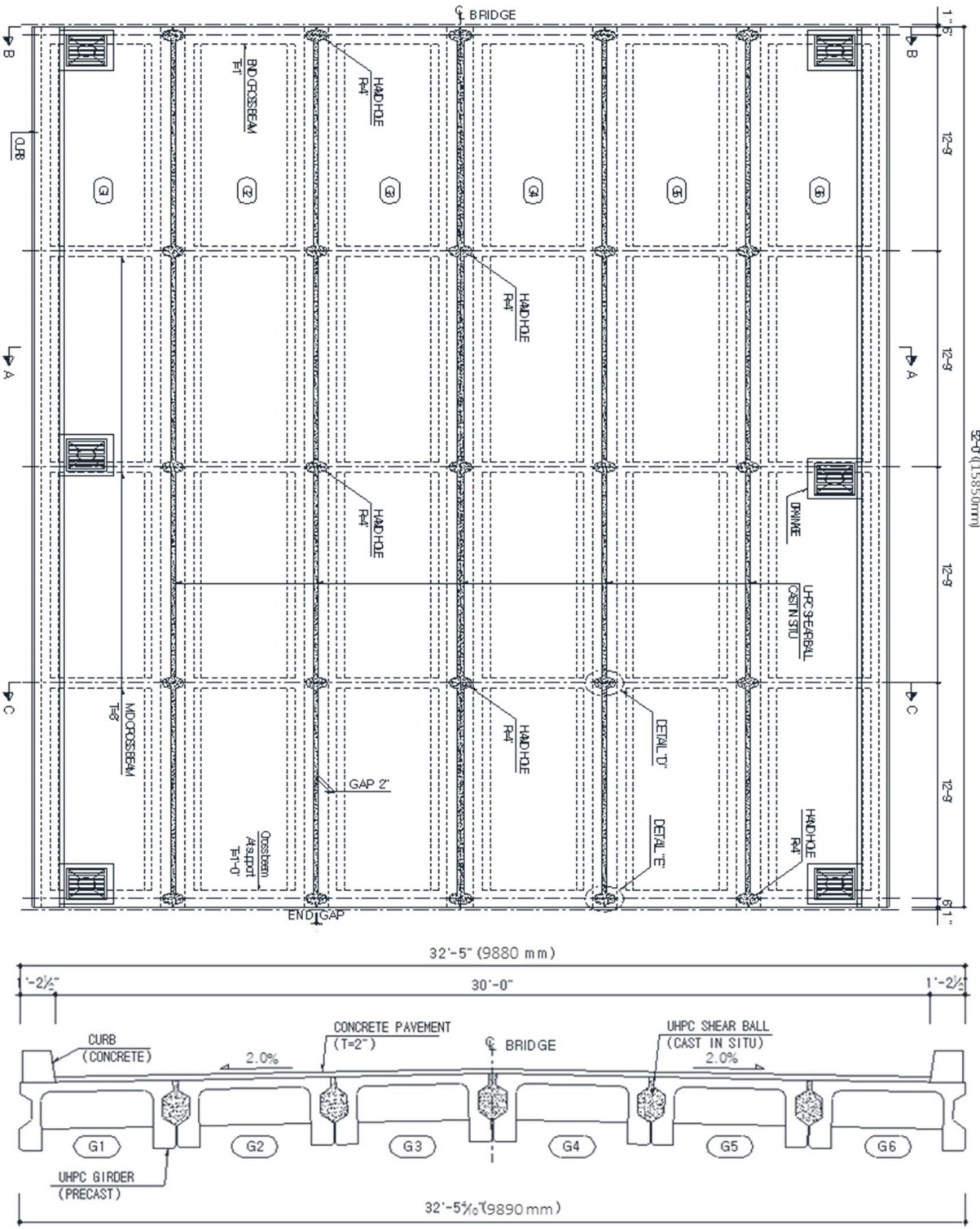

**Figure 5.** Plan view and cross-section of Hawkeye Bridge design.

The original pi-girder design was developed by the Massachusetts Institute of Technology (MIT, Cambridge, MA, USA) with a 33″ tall girder covered with a 3″ top-slab throughout the mid span and a 6″ thick slab at the ends of the girder [23]. The girders were prestressed, which allowed reinforcing steel to be eliminated. For the Hawkeye Bridge, the shape of the pi girders was optimized for UHPC to minimize the cross-section while exploiting the superior properties of UHPC materials. The very high tensile and compressive strengths of UHPC allowed a thinner slab and slimmer girders, making it possible to combine slab and girder in a single piece. Compared to the MIT's pi-girder design, as shown in Figure 6, the height of a UHPC girder for the Hawkeye Bridge was reduced from 33″ to 28″ while the thickness of the slab was increased from 4″ to 4.5″. Each girder was post-tensioned rather than prestressed. To provide a more stable structure, UHPC girders were laterally post-tensioned at the jobsite through the crossbeams.

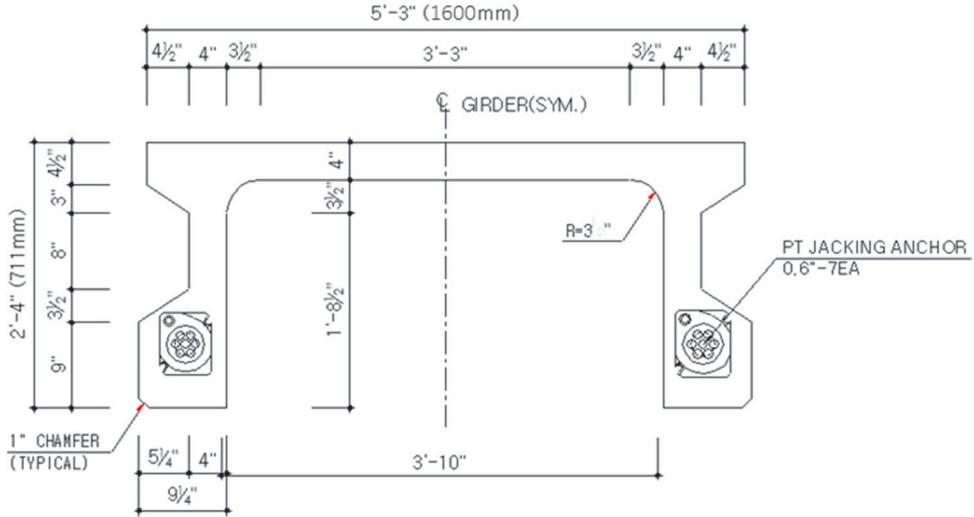

**Figure 6.** Hawkeye Bridge pi-girder cross-section design.

## 5. Construction of the Hawkeye UHPC Bridge

As shown in Figure 7, after demolishing the unstable existing bridge, construction of the substructure of the new UHPC bridge started in May 2015. First, six H-shaped steel piles were driven into the soil, approximately 12 ft (3.7 m) deep. Concrete stub abutments were constructed on top of the pile foundation. All pictures in this section were captured by the authors.

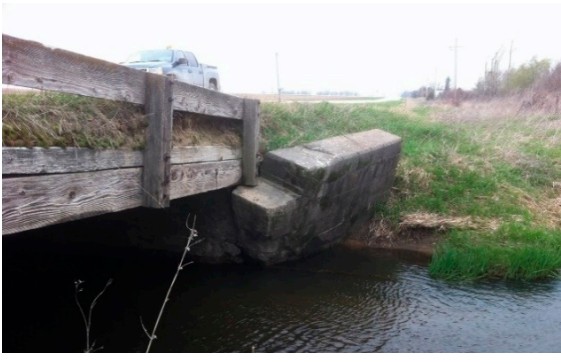 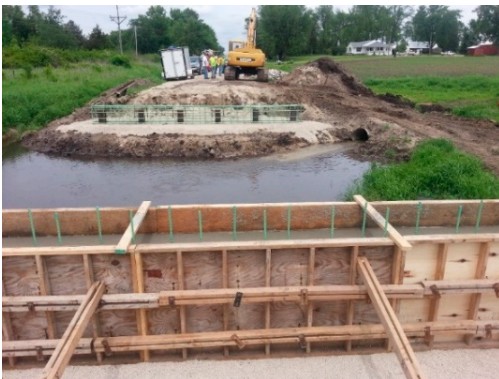

**Figure 7.** Existing unstable bridge (**left**) and construction of substructure of new UHPC bridge (**right**).

On 23 June 2015, construction of the first UHPC girder commenced at the yard of Buchanan County Office, Iowa. Due to dry, hot weather conditions, an additional 10% of superplasticizer was added to the mix to increase the workability. Table 3 summarizes the final mix design for the ready-mix UHPC along with mixing instructions in the field. Initially, we added cement to the wet sand and it created many cement balls because of the quick hydration of cement by the moisture in the wet sand. To prevent the creation of cement balls, we mixed cement with less reactive premix and then added the wet sand in a ready-mix truck.

**Table 3.** Mixing proportions and mixing instructions for UHPC.

| Mixing Orders | UHPC MIX | Total (lb/5.5 CY) | Mixing Instruction |
|:---:|:---:|:---:|:---:|
| 1 | Premix | 4386 | |
| 2 | Cement | 7310 | Mix for 10 min |
| 3 | Wet sand (MC = 4.2%) | 8041 | Mix for 5 min |
| 4 | Water | 1710 | Rotate at 10 RPM |
| 5 | Shrinkage reducer | 73 | Mix for 5 min |
| 6 | Defoamer | 5 | Add at 10 RPM |
| 7 | Superplasticizer | 140 | Mix for 5 min at maximum speed |
| 8 | Steel fiber (0.63 inch long) | 362 | Add for 20 min at 10 RPM |
| 9 | Steel fiber (0.78 inch long) | 723 | Mix for 2 min at maximum speed |

For the first batch, bolts at the bottom of formwork were pulled out of the form due to the lateral pressure and leaking concrete mix. Therefore, as shown in Figure 8a, a special wooden girder form was fabricated to ensure the strength needed to resist the lateral pressure generated by the weight of the flowable UHPC mix. It took 11 cubic yards (8.4 m$^3$) of UHPC mix to fill a form, where each ready-mix truck produced 5.5 cubic yards. Compared to the laboratory mix, only 82% superplasticizer was needed for the ready-mix truck due to its better mixing capability. As shown in Figure 8b, steel fibers were delivered to the ready-mix truck using a vibrating mesh and a conveyor belt. As shown in Figure 8c, flowable UHPC mix was poured into a form and the surface was leveled. Since the flowable UHPC mix set very quickly as it came out of the ready-mix truck, another batch of mix had to be discarded. After removing the formwork, the UHPC girder was steam-cured by placing heated water hoses to provide sufficient heat and water with a gradual heat application rate set at 10–15 °C/hr. As shown in Figure 8d, each girder was post-tensioned using seven 0.6″ (1.5 cm) diameter longitudinal strands with a total force of 300 kips (2.67 MN) to achieve a gauge pressure of 6500 psi (44.8 MPa) at both bottom ends. Strands were pulled up to 80% of the ultimate tensile strength, resulting in a 6 inch elongation. On 9 September 2015, six steam-cured girders were installed at the jobsite. As shown in Figure 8e, each of five transverse crossbeams was post-tensioned using three 0.6″ (1.5 cm) diameter strands with a total force of 105.3 kips (0.468 MN), where 35.1 kips (0.156 MN) was applied to each strand to achieve a gauge pressure of 4500 psi (31.0 MPa). Once all three strands were post-tensioned, a duct was grouted and a cap was installed. As shown in Figure 8f, the unique Hawkeye Bridge was successfully constructed using innovative UHPC materials.

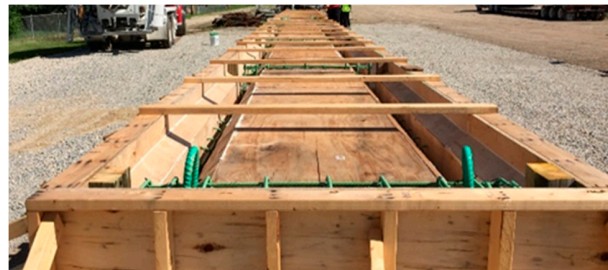

(**a**) Girder formwork

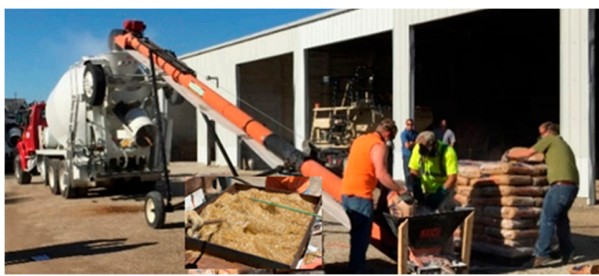

(**b**) Adding premix/steel fibers to ready-mix truck

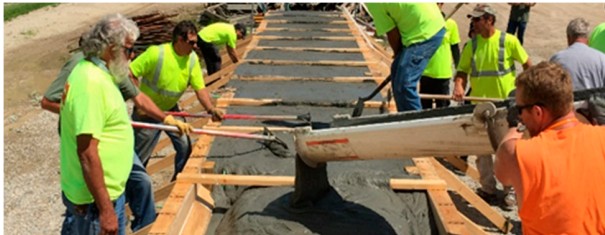

(**c**) UHPC pouring

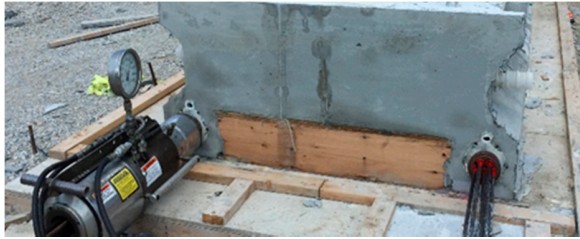

(**d**) Longitudinal post-tensioning of girder

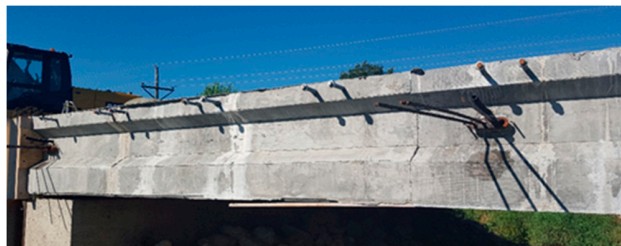

(**e**) Transverse post-tensioning of crossbeam

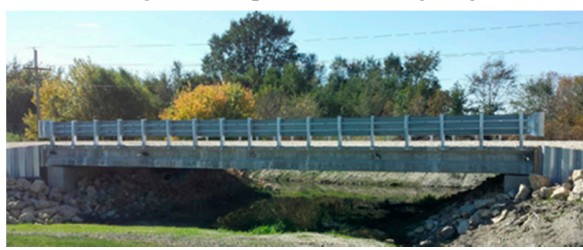

(**f**) Completed Hawkeye Bridge

**Figure 8.** UHPC mixing, pouring, and construction process of Hawkeye Bridge.

## 6. Monitoring of Bridge Deck Joints

As shown in Figure 9, each SenSpot strain sensor consists of two parts: a strain gauge with a precision of 1 microstrain and a wireless transmitter. The wireless transmitter converts analog strain measurements from the strain gauge to digitized data and sends them wirelessly to the remote transmission gateway, which then sends the data to a remote cloud server through a cellular service. The wireless transmitter can transmit digital strain data every six seconds or at longer intervals. The sensors are expected to last for a minimum of 10 years without a battery replacement.

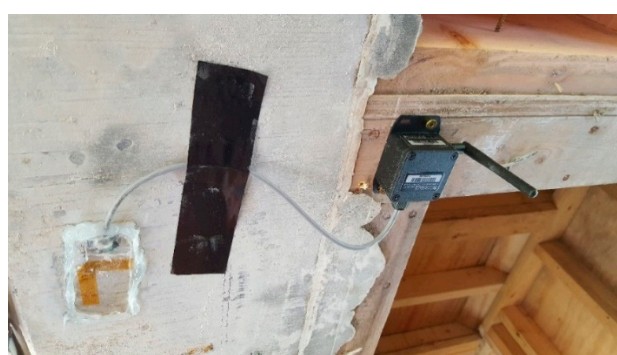

**Figure 9.** Installed strain gauge and wireless transmitter.

### 6.1. Installation of Strain Gauges

As shown in Figure 10a, the gaps between precast girders were stuffed with rubber pad and shear bulbs were then filled UHPC materials. To monitor the short-term performance of the bridge deck joints, as shown in Figure 10b, six sensors were installed at the left and right sides (L and R) of three joints from the edge to the center of the bridge (1, 2, and 3). Two strain sensors were attached at the bottom of each of three joints using a superglue. Silicon was then applied around the strain gauges for protection from moisture.

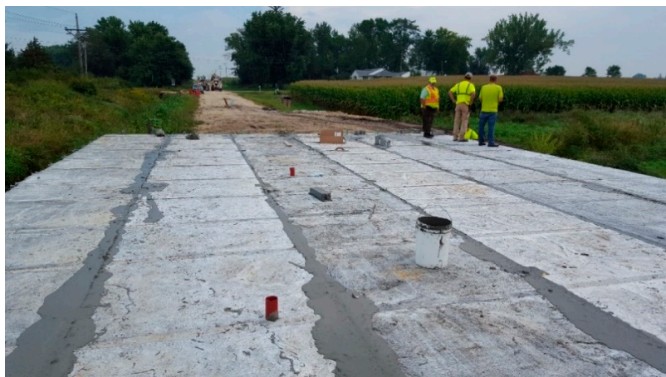

(**a**) Bridge joints filled with UHPC materials

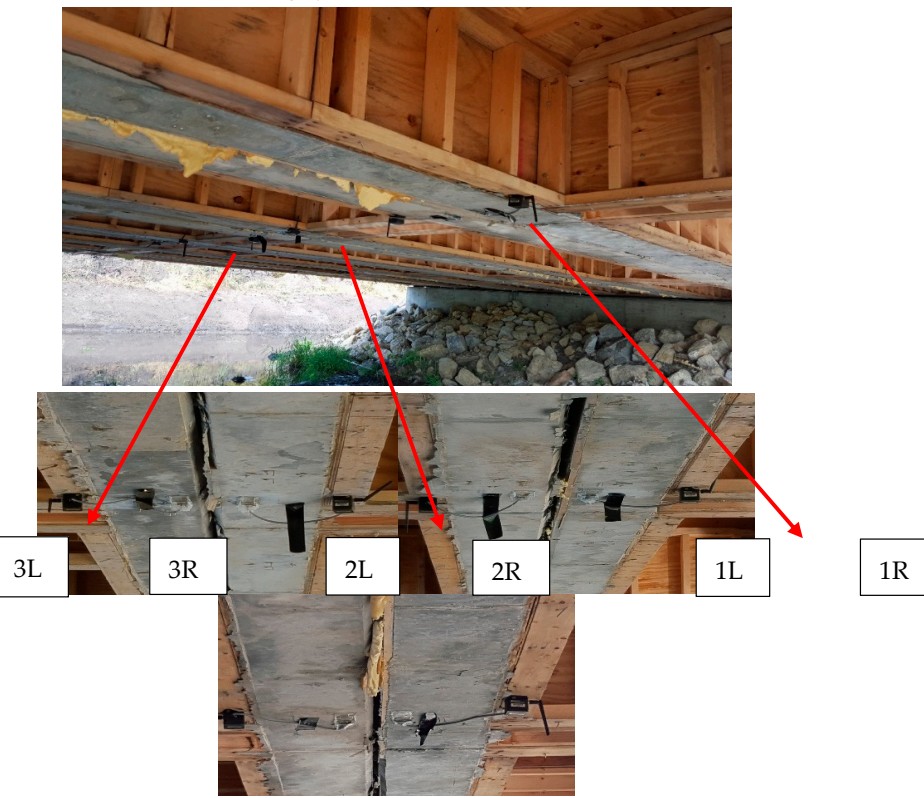

(**b**) SenSpot strain sensors installed on three joints from edge (1L, 1R, 2L, 2R, 3L, and 3R)

**Figure 10.** Six SenSpot strain sensors installed on three joints.

### 6.2. Loading Tests

Loading tests were performed using a standard tandem-axial dump truck adopted by Buchanan County, Iowa, which is 26′-8″ (8.1 m) long and 8′ (2.4 m) wide with a wheel base of 18′-8″ (5.7 m) and a tandem axle spacing of 4′-6″ (1.4 m). The loading truck was driven at a crawl speed of 2–3 mph. As shown in Figure 11, the gross weight of the loading truck was 50,000 lb (22.6 ton) with a rear tandem axle load of 40,000 lb (18.1 ton) and a front single axle load of 10,000 lb (4.5 ton). The dump truck was driven on top of each joint in

such a way that the rightmost wheels were placed on top of a joint. First, the truck was driven 10 times on top of the rightmost joint. The truck was then moved laterally by about six feet and driven 10 times on top of the second joint; this was repeated on the third joint located at the center of the bridge.

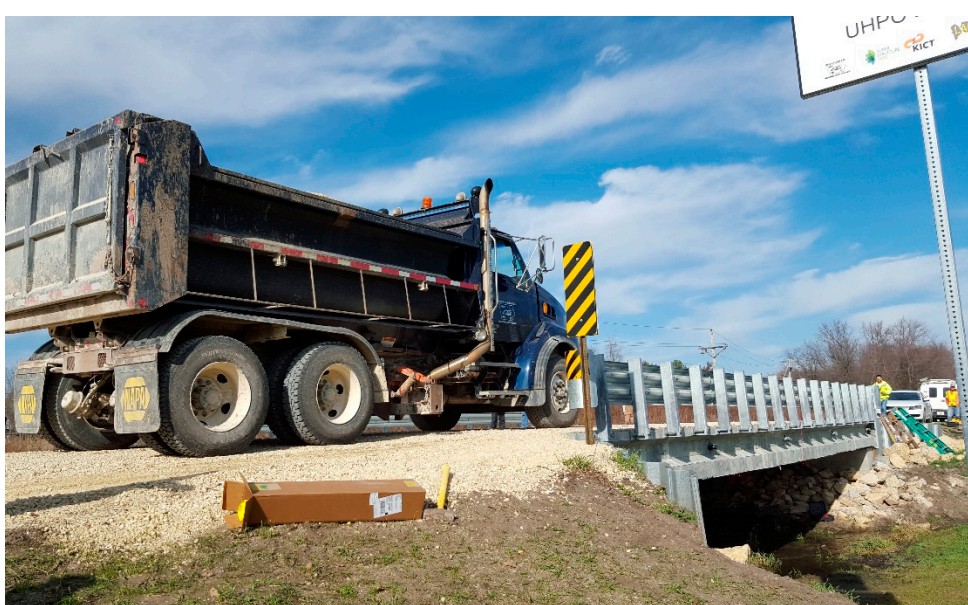

**Figure 11.** A heavy tandem-axial dump truck loading on bridge joints.

To evaluate the short-term performance of the bridge joints after construction, strain data were collected from both sides of each joint. Figure 12a shows example strain data collected from the left and right sides of joint 3 two years after construction. It shows three sets of strain data collected on zero, two and four years after construction while a truck was driven on top of the joint. A close-up view of the third loading set is shown in Figure 12b, where the highest strain value from sensors 19 and 21 were 10 and 13 microstrains, respectively. The highest strain value from each sensor was selected to represent peak loading when the truck was placed right on top of the joint.

Figure 13 shows a plot of strain data collected from three joints at three time-points: (1) right after construction, (2) two years after construction, and (3) four years after construction. As can be seen from Figure 13, the first loading test right after construction in 2015 resulted in relatively high strain values, particularly the 36 microstrains at joint 1. It can be postulated that these high strain values were due to not well-seated bridge girders and incomplete curing of the UHPC filling materials at the joints. However, it should be noted that strain data collected from the left and right sides of each joint were quite similar. Strain data collected in the second and the fourth years after construction were lower, ranging between 10 and 20 microstrains. Overall, the two sensors installed on both sides of the same joint exhibited similar peak strains, which confirmed that the stresses applied to the joint were evenly distributed across the joint.

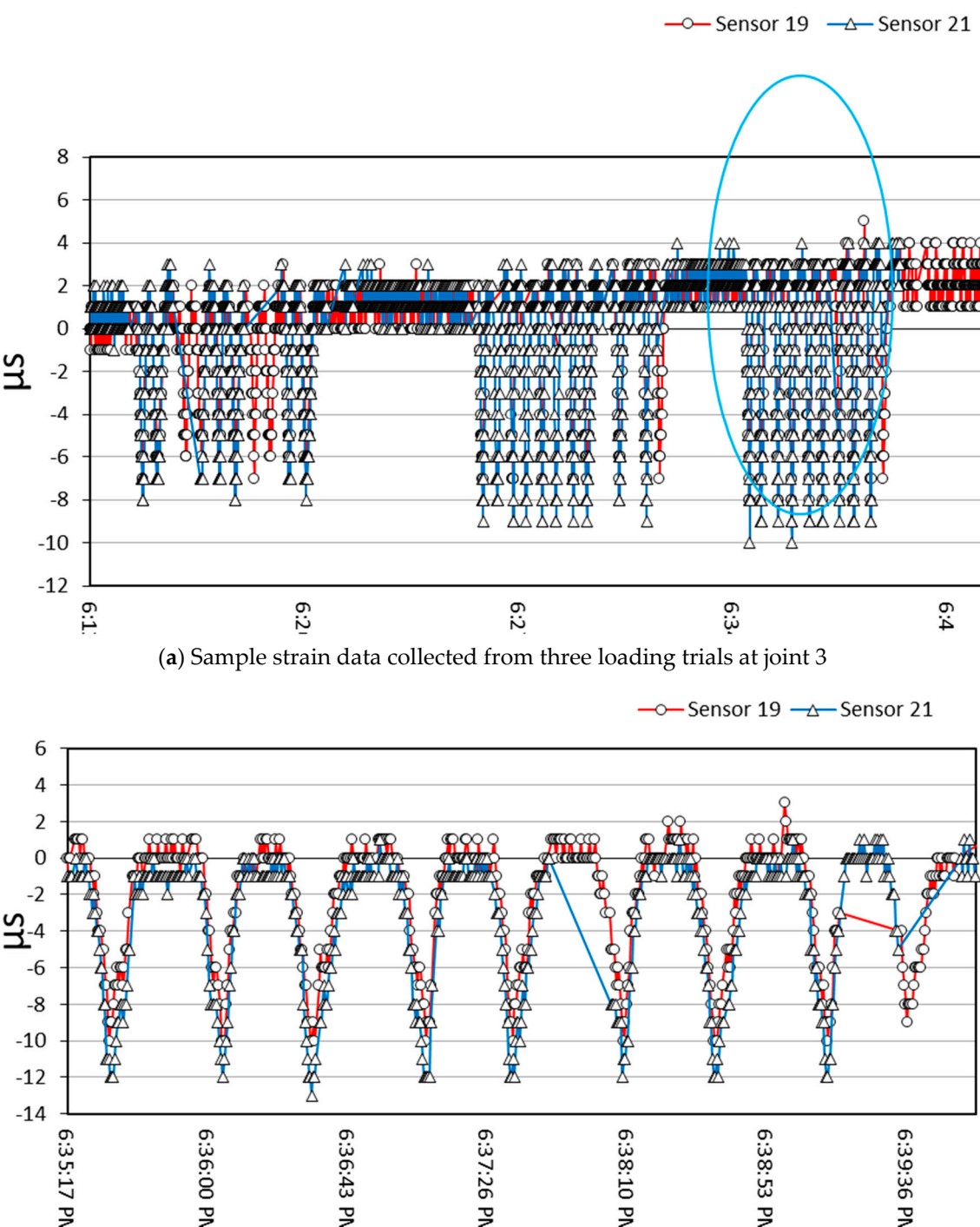

(**a**) Sample strain data collected from three loading trials at joint 3

(**b**) Detailed sample strain data collected from joint 3 on the third loading trial

**Figure 12.** Example strain data collected from joint 3 two years after construction.

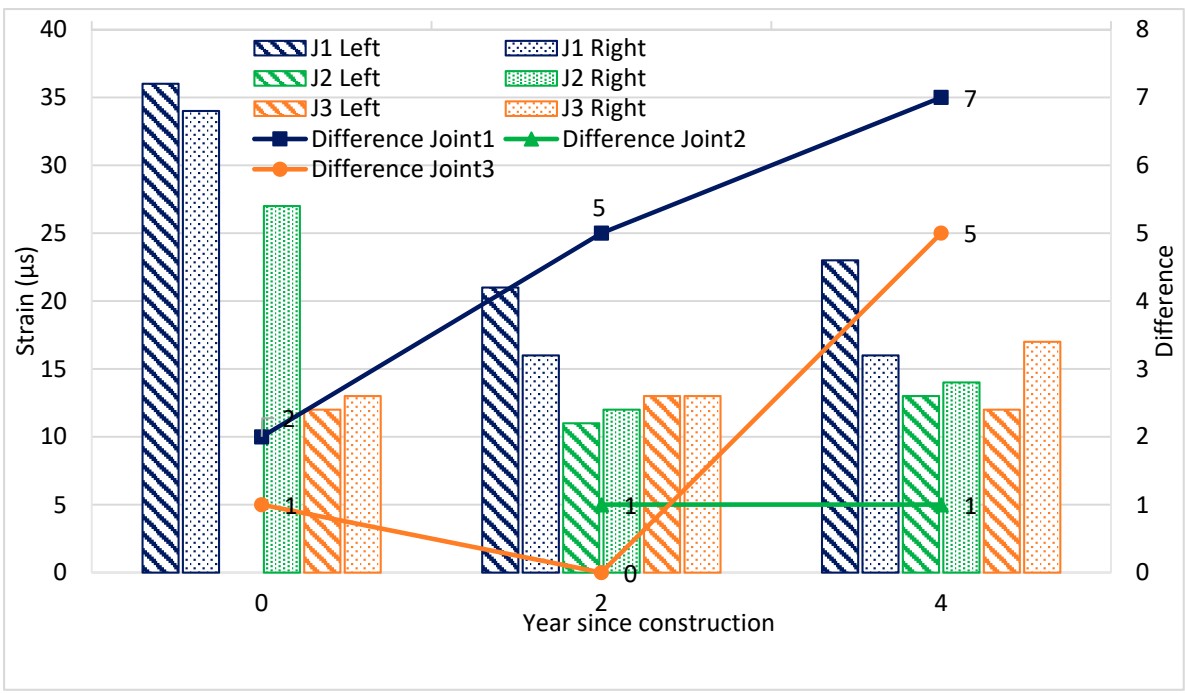

**Figure 13.** Strain data collected from left and right sides of each of three joints at 0, 2, and 4 years after construction.

## 7. Summary and Conclusions

In the past 15 years, UHPC materials have been applied in over 250 bridges in the US, mostly as precast concrete deck panels and field-cast connections between prefabricated bridge elements. This paper presents the laboratory test results, design, field construction, and monitoring of a new bridge using a unique UHPC technology which utilizes two different lengths of steel fibers of 16.3 mm and 19.5 mm at a 1:2 ratio by weight. Lengths of steel fibers were optimized to increase both tensile strength and the toughness. This unique UHPC material was used to build a pi-girder bridge in Buchanan County, Iowa. After the bridge was constructed, the bridge joints were monitored three times over four years using sensors attached to both sides of each of three joints.

First, laboratory testing of the UHPC material was performed to determine compressive and indirect tensile strengths. The compressive strength satisfied the target compressive strength of 200 MPa (29,000 psi) after 28 days, with a very significant early gain in the compressive strength of 87.9 MPa (12,750 psi) in one day. The indirect tensile strength of UHPC materials achieved 18.3 MPa (2654 psi) in 7 days, which is significantly higher than that of regular concretes.

A pi-girder design with a box-shaped joint between girders was adopted to reduce construction cost and protect the post-tensioned bottom flange from potential environmental damage. The proposed UHPC material was produced with local cement and aggregates using a regular ready-mix truck. Flowable ready-mix UHPC materials were poured into a prefabricated form and a total of six UHPC girders were steam-cured using heated water hoses at the Buchanan County yard. All precast pi-girders were post-tensioned using longitudinal strands at both bottom ends. Six precast UHPC girders were installed at the jobsite using a crane and laterally post-tensioned through the crossbeams.

During the construction process, we learned the following lessons through wasting batches of UHPC mixes:

- We observed many cement balls due to quick hydration of cement with wet cement. Because UHPC does not use coarse aggregates, the chance of forming cement balls is very high. We solved the cement ball problem by separating cement from wet sand by mixing cement with less reactive premix first before adding wet sand to the ready-mix truck.

- Bolts used to tie wood formwork panels at the bottom were separated due to the lateral pressure from the flowable UHPC mix. We solved this problem by doubling the number of bolts at the bottom of the formwork.
- Due to the use of a high amount of superplasticizer with minimum water, there was a very narrow time window between flowing and setting of UHPC mix. Therefore, we optimized the mixing time so that UHPC mix did not start setting in the ready-mix truck.

To evaluate a short-term performance of the bridge joints after construction, strain data were collected from each of three joints. High strain values were observed right after construction due to unsettled bridge girders and incomplete curing of the UHPC filling materials at the joints. However, strain data collected from the left and right sides of each joint were very similar. Strain data collected in the second and fourth years after construction were lower, ranging between 10 and 20 microstrains, and the differences between strains from left and right sides of each joint were 7 microstrains or less. Based on similar peak strains from both sides of each joint four years after construction, it can be concluded that the joints between girders have been performing well. Hawkeye bridge was successfully constructed utilizing innovative UHPC materials and has been performing satisfactorily.

The adoption of UHPC materials in rehabilitating bridges and constructing primary bridge components is progressing rapidly in the world because of UHPC's unique characteristics of the higher strength and longer durability than regular concretes. This innovative slender and longer girder design is aesthetically pleasing and provides more open space, which should be a preferred bridge design in smart cities.

**Author Contributions:** The authors confirm contribution to the paper as follows: study conception and design: H.L., B.-S.K. and B.K., data collection, processing and analysis: H.K., X.H., B.M., draft manuscript preparation: H.K., X.H., B.M., H.L., laboratory tests: H.K., G.-S.R., K.-T.K., field construction and documentation: H.K., C.J., B.-S.K. and B.K. All authors have read and agreed to the published version of the manuscript.

**Funding:** This research received an external funding from Korea Agency for Infrastructure Technology Advancement (KAIA).

**Institutional Review Board Statement:** Not applicable.

**Informed Consent Statement:** Not applicable.

**Data Availability Statement:** No new data were created or analyzed in this study. Data sharing is not applicable to this article.

**Conflicts of Interest:** The authors declare no conflict of interest.

**Disclaimer Notice:** The contents of this paper reflect the views of the authors, who are responsible for the facts and the accuracy of the information presented herein. The opinions, findings, and conclusions expressed in this paper are those of the authors and not necessarily those of the sponsors. The sponsors assume no liability for the contents or use of the information contained in this document.

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
