# Peer review of "Construction and Performance Monitoring of Innovative Ultra-High-Performance Concrete Bridge"

_infrastructures, doi:10.3390/infrastructures6090121_

Round 1

Reviewer 1 Report

The paper is interesting. Some comments can be found below:

- The introduction is unacceptably too short.

- We live now in a climate emergency so its most strange that the authors have not start the paper by mentioning exactly that. It seems that they are not aware about the words of a Professor of Physics at the University of Oxford authored a paper where one can read the following:

 “Let’s get this on the table right away, without mincing words. With regard to the climate crisis, yes, it’s time to panic”

Pierrehumbert, R., 2019. There is no Plan B for dealing with the climate crisis. Bulletin of the Atomic Scientists, pp.1-7.

So please start the introduction by draw a connection between environmental degradation, resource efficiency, and concrete durability

- The introduction lacks a rationale for the paper

- In lines 60-61 the authors refer to “the high cost of steel fibers in UHPC” that is scarce. The authors need to add to more data on cost so readers can have a minimum information to allow comparisons between normal concrete and ultra high performance concrete bridge

-How did authors have calculate the concrete composition ?

Reviewer 2 Report

The manuscript entitled "Construction and Performance Monitoring of Innovative Ultra High Performance Concrete Bridge” presented the design, construction, and performance monitoring of a new bridge constructed in Iowa using a unique UHPC technology. The concrete mix included steel fibers with different lengths.

The manuscript lacks clarity and should not be accepted in this current condition. The fluent of the manuscript is not clear to this reviewer. This reviewer recommends major editing and resubmitted for re-review.

Technical comments:

  1. The abstract should be re-organized to be in this order: introduction to the problem, methodology used, measurements, and conclusion.
  2. The manuscript is very poor in writing and could benefit greatly from professional editing to improve technical writing and English.
  3. Section 2: The authors presented common information about UHPC and its applications. They should highlight what is the problem. Therefore, the literature review is poor. The authors should increase their discussion on previous related research and highlight how their study is providing a different approach or adding significantly to what has been done.
  4. The information provided in lines 92-108 should be titled differently like "Material and Specimens Preparation".
  5. More details about the prepared samples and curing process should be provided.
  6. What is the relation between these designed mixes and the constructed bridge? More clarification should be added to the manuscript. Also, the main purpose of section 3 should be highlighted.
  7. The information in Table 1 for which mix? The authors presented three different mixes in line 104.
  8. Line 106: The authors should clarify who determined these extra amounts to improve or maintain workability.
  9. Cannot see any standard deviations in Figures 3 and 4.
  10. Table 3: Why the other results are missing? Also, why did the authors choose day 4 to conduct the compression tests? Typically, it should be after 3 days (1, 3, 7, 14, and 28).
  11. The plan shown in Figure 5 is not clear at all.
  12. Line 144: How much is the post-tensioning force as a percentage of the ultimate strength of the tendons? According to what standard this bridge was designed? More details should be added.
  13. Section 6: This section is lacking discussions. In detailed discussions should be provided to clarify the recorded data.
  14. Line 240: What are the dimensions of this truck? Is it consistent with the standards? Which standard?
  15. Line 272: What do you mean by "in two years"? I think you mean after two years. or Do you mean data during the two years?

Reviewer 3 Report

This is an excellent high-performance concrete related manuscript, focusing on the case study of a bridge, it is suitable for publication in this journal as long as some considerations are addressed, such as:

a) Correct in line 1 "eArticle" by "Article"
b) The title I would suggest adding that this is a case study;
c) The abstract, and other parts of the text, should be revised in terms of formatting, for example, adding the text in a justified way, in addition to adding some quantitative data from the case study;
d) In the literature review, authors should implement some adjustments and insert some current research on cementitious materials, showing their technical properties, I suggest the following works: 10.1016/j.cscm.2020.e00406; 10.1016/j.cscm.2021.e00604; 10.1016/j.conbuildmat.2020.118786; 10.1016/j.cscm.2021.e00583; 10.1016/j.jmrt.2020.03.122; 10.1016/j.jobe.2021.102662.
e) In table 1 the units must be revised, do not use "^";
f) In figure 3, the graph has superimposed numbers, check;
g) The results must be presented in the SI, consider MPa, for example;
h) Are the case study images from the authors themselves? Indicate and declare it!

Round 2

Reviewer 2 Report

The authors have addressed most of the reviewer's comments. However, the authors should clear the word track to make the manuscript clear for the reviewer. Just highlight the modifications or the new text.

Reviewer 3 Report

The authors made all, or almost all, the suggested corrections, however I still think it is pertinent in the title to state that this is a case study, I leave this decision to the editor. Other comments were adequate.